# Non-Institutional Childbirths and the Associated Socio-Demographic Factors in Gambella Regional State, Ethiopia

**DOI:** 10.3390/ijerph18062859

**Published:** 2021-03-11

**Authors:** Gnkambo Agwa Gora, Muhammad Farooq Umer, Peter Obang Ojulu, Sintayehu Tsegaye Betaw, Akwoma Okugn Cham, Ojulu Agwa Gora, Xin Qi

**Affiliations:** 1Gambela People National Regional State Bureau of Health, Gambela People National Regional State, Gambella 5440, Ethiopia; goragnkambo@gmail.com; 2Department of Epidemiology and Biostatistics, School of Public Health, Xi’an Jiaotong University, Xi’an 710061, China; rafooq@hotmail.com; 3School of Public Health, Alshifa Trust, Rawalpindi 46600, Punjab, Pakistan; 4Department of Education, Gambella Teacher Education and Health Science College, Gambella 5440, Ethiopia; peter_obang@yahoo.com; 5Health Science College, Defense Force University, Bishoftu 1041, Ethiopia; sintayehutsegaye288@yahoo.com; 6Public Health Emergency, Gambella People National Regional State, Bureau of Health, Gambella 5440, Ethiopia; akwomaokogn@yahoo.com.au; 7Disease Prevention and Control, Gambella People National Regional State, Bureau of Health, Gambella 5440, Ethiopia; goraagwa@outlook.com; 8Global Health Institute, Xi’an Jiaotong University, Xi’an 710061, China

**Keywords:** non-institutional childbirth, institutional birth, socio-demographic factors, Gambella Regional State

## Abstract

The decades-long global efforts to reduce maternal morbidity and mortality have shown overall progress, but most developing countries are still lagging significantly. This study aimed to assess the prevalence of non-institutional childbirths in the Gambella State and to identify socio-demographic factors responsible for non-institutional utilization of available birth services by reproductive-aged mothers. A community-based cross-sectional study design was adopted using a multi-stage random sampling technique. Binary logistic regression was used to identify factors associated with the selected place of birth. EpiData version 3.1 and SPSS version 13.0 were applied for data entry and analyses. All the 657 eligible mothers recruited for this study responded to the interview. 71% of the total respondents had non-institutional childbirths (NICB), and the rest had their most recent childbirth in an institution with skilled healthcare providers’ assistance. Socio-demographic factors were significantly associated with NICB. Nuer (AOR = 2.12, 95% CI: 1.23–3.63) and Majang ethnic (AOR = 1.98, 95% CI: 1.02–3.83) groups had higher rates of NICB than the rest of the study population. The prevalence of non-institutional childbirths in Gambella remained two times higher than the institutional childbirths.

## 1. Background

The decades-long global efforts to reduce maternal morbidity and mortality have shown overall progress, but most developing countries are still lagging significantly [1]. Globally, the total number of maternal deaths has reduced markedly (from 1990 to 2013; it declined by 45% at an average annual rate of 3.3%) [2]. While in Sub-Saharan Africa (SSA), mothers’ risk of dying from preventable complications of pregnancy and birth is still very high, i.e., 1 in every 22 live births [1]. In 2015, an estimated 239 maternal deaths per 100,000 live births occurred in developing countries, and 66% of these maternal deaths took place in Sub-Saharan Africa (SSA), while only 12 per 100,000 live births occurred in developed countries [1]. These statistics suggested that developing countries call for exceptional focus if the United Nation’s goals of bringing down global maternal mortality rates to under 70 per 100,000 live births are to be attained by 2030 and the maintenance of uniform better maternal health, universally [3].

The top causes of maternal mortality in developing countries have been bleeding and complications resulting from childbirth [4]; in SSA, blood loss during childbirth alone accounts for one-third of all the maternal deaths [5]. These maternal and newborn deaths can be avoided by simple interventions like conducting childbirths at healthcare facilities that would enable childbearing women to receive proper medical attention and care [6]. There is enough data to believe that skilled birth attendance is the most significant intervention to prevent life-threatening complications of pregnancy and childbirth [7]. Not surprisingly, the rate of institutional childbirths (ICBs) in Africa and Southern Asia has been the lowermost in the world (estimated to be around 50%) [7]. Moreover, there exist wide disparities in institutional and non-institutional childbirth (NICB) practices within the developing countries between the affluent and the poorer communities. Poor mothers in remote areas are least likely to receive adequate healthcare than those dwelling in the wealthiest households (43% versus 89%) [8].

Ethiopia is amongst the countries with the highest maternal mortality rates with an assessment of 353 and 412 per 100,000 live births [9]; in 2013, it had the highest maternal mortality rate worldwide [10]. In Ethiopia, around 2.9 million women give birth annually, over 25,000 die, and more than 500,000 suffer from severe impairment such as obstetric fistulas [11,12], mainly due to lack of maternal healthcare. This situation is worsened by the higher national population growth rate, gender inequality, early marriages, female genital mutilation, unplanned pregnancies, illegal abortions, and sexually transmitted diseases, including HIV/AIDS [13,14]—causing stressful consequences like social, emotional, and economic disasters [12].

Socio-demographic factors, e.g., the distance to a healthcare facility, the frequency of antenatal care, knowledge on pregnancy and childbirth danger signs, history of prolonged labor, and decision-making status, are among the major factors cited for low acceptance of ICB [15]. Government is, however, trying hard in its limited resources and seems committed to enhancing the Reproductive Health (R.H.) status of women, men, and young people of Ethiopia through family planning, maternal healthcare, newborn healthcare, and socio-cultural health activities [13,14,16]. The studies addressing determinants of non-institutional childbirths in Gambella are limited. The available literature predominantly discussed urban areas and healthcare centers only and provided scarce information on rural areas.

Furthermore, no available research described a holistic picture encompassing all the districts in the Gambella Regional State. Therefore, it is crucial to identify and describe factors associated with non-institutional childbirths and institutional services’ underutilization across the regional state. This study explores the prevalence of NICBs in Gambella and figures out the socio-demographic variables related to the utilization of available birth services among reproductive-aged mothers.

## 2. Materials and Methods

### 2.1. Study design

Mothers aged 15–49 years in selected districts of Gambella who gave birth within 2 years before the study were included by a multi-stage simple random sampling technique. We set up the inclusion and exclusion criteria of selection. Inclusion criteria: All mothers who delivered a child less than 2 years ago or all mothers within 23 months of post-delivery with a live child, all districts with functional healthcare service delivery points, mothers who were mentally and physically capable of being interviewed, permanent resident of the study area (minimum of 1 year), and living near a functional healthcare facility in selected districts of the Gambella region and chosen by simple random sampling technique. Exclusion criteria: The refugee population in the region, communities with dysfunctional or breakdown of healthcare service delivery points, mothers who were critically ill, could not talk or listen, households without eligible mothers, and mothers from a neighboring or bordering region during the time of data collection. A community-based cross-sectional study design employed quantitative data collection methods to assess NICB service utilization factors in Gambella using a structured questionnaire. The adapted and structured questionnaire developed from different similar World Health Organization (WHO) literature in English and translated into ‘Amharic’ language (local language) consisted of 26 categorical variables. The actual sample size was determined using a single population proportion equation [*n* = [(Z α/2)^2^ p (1 − p)/d^2^]. Estimating unskilled birth prevalence in Gambella as 50% with a 95% level of significance and allowing a 5% margin of error population size for women of reproductive age yielded a sample size of 597 plus 10% for non-respondents, making the total calculated sample size 657. Among the 14 districts in Gambella, 13 districts were selected based on eligibility criteria to represent the entire region for this study. Only the Akobo district was excluded from the study due to its absence of a healthcare service delivery point. After the requirement of a sample size of 657 was calculated, the samples were distributed to 13 districts according to the population weight. Based on the selected 13 districts, 5 villages from each district were selected based on the accessibility of healthcare services from the village. Then, 65 villages were included in the study to get 657 study subjects. A convenient sampling technique was used to select 5 villages in each district by considering the estimated number of deliveries in the village population. Finally, study participants were chosen using a random sampling technique. A systematic cluster sampling method was used to select the study subjects. Eight data collectors who spoke the local language, with an education level of Grade 12 and above, were hired. Four supervisors who had a master’s degree were picked from Gambella Regional Bureau of Health, Districts Health Offices, and Health Centers. Five days of training were given to data collectors and supervisors on the research’s objective, data collection procedures, and how to keep confidentiality. Team members were instructed to check the completeness of each record at the end of each interview. Then, the data were collected using house to house and face-to-face interview questions. A maximal effort was made to ensure privacy during the interview. Supervisors were with data collectors every day to support data collectors and check data daily for completeness, accuracy, clearness, and consistency to make necessary corrections. The investigator rechecked the file’s completeness immediately during submission, made changes, and edited and coded the supervisor’s results. Data were collected from January to February 2018. Data cleaning was done through the double-entry to Epi Data version 3.1 software package (Epidata Association, Denmark).

### 2.2. Study Area

This study was conducted in the Gambella People Nationa Regional State Southwest of Ethiopia, at the community level, from December 2017 to December 2018. Gambella is the Capital City of Gambella Peoples National Regional State, covering 23,127 sq. km. Gambella town is located 767 km from Addis Ababa, Ethiopia’s capital city, in the southwest direction. According to the 2007 National Census Agency (CSA), the region had a total population of 307,096 [17]. Based on the 2016 projected total population of the area was about 422,002 people, and the entire community of the capital town was 42,812. Administratively, the regional state was divided into 3 zones and 1 special district, which were furtherly sub-divided into 14 districts. The regional official working language was the Amharic language. Different ethnic groups were residing in the region, including 5 major ethnic groups (Anywaa, Majang, Komo, Opuo, and Nuer) and other ethnic groups, including Oromo, Kembata, Gurage, Tigre, Amhara, and population from different parts of the country and refugees from South Sudan. In this study, age, marital status, occupation, educational status, religion, monthly income, and ethnicity were independent variables. The dependent/outcome variable was the place of birth in the last 2 years before this study’s survey. This variable was binary (institutional delivery or non-institutional delivery).

### 2.3. Data Analysis

The corresponding code number was written carefully at each margin. The template scheme for data entry was developed and pre-tested for ranges, skip patterns, and allowed correct values by entering 30 questionnaires. After this validation, the principal investigator entered the data using Epi Data version 3.1 and exported it to SPSS 13.0 version (SPSS Inc. Chicago, IL, United States) statistical software packages for data analysis. Compute frequencies and summary statistics were used to describe the study population about relevant variables and outlines. Any errors identified were corrected after revising the original data using the code numbers and statistical commands. Frequencies and Chi-Square measures of variation were used to describe the study population about socio-demographic and statistically significant to relevant variables. Binary logistic regression analysis was completed using SPSS 13.0 version statistical software packages. The degree of association between independent and outcome variables was assessed using the crude odds ratio and adjusted odds ratio with a 95% confidence interval to control the potential confounding variables. Binary logistic regression was considered a more appropriate statistical method because the dependent variable was categorical. Calculating the odds ratio measured the strength of association, 95% CI, and *p*-values less than 0.05 for statistical significance with the determinant factors assessed.

## 3. Results

### 3.1. Socio-Demographic Characteristics of the Respondents

In this study, 657 women respondents were interviewed from 13 districts and 65 villages of Gambella. All eligible women in the selected districts responded to the questionnaires. The overall response rate was 100% (657). Out of the total respondents, over 80% of the total were in the age group of 15–39 years old. The majority were married; others were single or in another type of marriage. Regarding ethnicity, the majority were Anywaa, 235 (35.8%), followed by Nuer, 226 (34.4%), ethnic groups. Out of the 657 women, 468 (71.0%) had NICBs and others had institutional births. Of the 468 respondents who had NICB, 198 (72.0%) were in the age group of 30–39 years, and 80 (31.0%) of the total 189 institutional birth were in the age group of 15–29 years. Concerning the ethnicity of those who had NICBs, 178 (78.0%) were in the Nuer group, and 67 (77.0%) were in the Majang ethnic groups (Table 1).

### 3.2. Proportion of Respondent’s Education Status by Age Groups

Educational status was significantly associated with non-institutional childbirth compared to other factors. Mothers who had no formal education were almost three times more likely to attend NICB than mothers with a diploma or greater education level. The older age contributed significantly to non-institutional childbirth. Out of the total respondents, 33.3% of Grade 9–12 were in the age groups of 15–29 years and 23.6% in 30–39 years. The large proportion of no formal education, 49.2%, were 40–49 years. For adults aged 30–39 years, mothers with a diploma and above were 15.3%, which was lower than no formal education and Grade 5–8 groups but higher than Grade 1–4 groups (Figure 1).

### 3.3. Factors Facilitated Mothers to Pursue Non-Institutional Childbirths

Among the factors that contributed to non-institutional delivery, 37% of non-institutional birth was the distance from the residence of respondents from the healthcare facility. 36% was a fear of a male midwife or male nurse conducting childbirth service in the rural healthcare facilities, and 12.0% was a delay in ambulance service for transporting the mothers’ to at least the nearest healthcare facility (Figure 2).

### 3.4. Plan for the Place of Delivery Service Utilization

For mothers who replied “yes” for having a plan of the place of birth, 230 (59%) were non-institutional childbirth, and 160 (41.0%) were institutional delivery. Of those who replied “no” to have a plan of the place of childbirth, 238 (89.1%) were non-institutional childbirth, and 29 (10.9%) were institutional childbirth. Mothers who planned to have an institutional birth, 47.9%, were in institutional childbirth, and more than half, 52.1%, were non-institutional birth. However, for those who planned for non-institutional birth, almost 2% had their childbirth in institutional childbirth, and all 98% were non-institutional childbirth (Table 2).

### 3.5. Factors Associated with Non-Institutional Childbirths and Socio-Demographic Characteristics

The respondents’ marital status, occupation, education level, and ethnicity showed a statistically significant association with births in the logistic regression model. Mother’s level of education and ethnicity showed a statistically significant association with place of birth service utilization. Mothers who had no formal education were almost three times more likely to attend NICB than mothers with a diploma and above education level. Mothers who were Nuer and Majang by their ethnic group were about two times more likely to attend NICB than mothers from other ethnic groups in the region. However, the respondents’ age, religion, and monthly income did not show a statistically significant association with birthplace based on the logistic regression model output (Table 3)

## 4. Discussion

The study discussed women’s childbirth practices and associated socio-demographic factors across all Gambella districts, including rural and urban areas. Our analysis also examined many of the critical factors associated with the under-utilization of ICBs. This study provided novel information on childbirth practices in the state, which can add valuable information on the healthcare delivery point distribution in rural areas for public health managers and implementers to improve maternal and child health in the country regional states. In this study, around two-thirds of the mothers had non-institutional childbirths, while only one-third had institutional childbirths with modern healthcare providers’ assistance. These statistics showed that many mothers are giving birth non-institutionally, against the World Health Organizations’ recommendations, as stated in Sustainable Development Goals (SDGs) [18]. Our study results do not follow the data shown by the Ethiopian Demographic and Health Survey 2016 (EDHS 2016) for the Gambella State, which showed that institutional childbirths were around 45% [9]. The findings are also not in agreement with a previous study conducted on institutional childbirths in the Amara region, Bahir Dar City, demonstrating 21.2% non-institutional and 78.8% institutional births. Another survey from the Bench Maji zone, Southern Nation Nationality Regional State, indicated that 21.7% and 78.3% were non-institutional and institutional births [19,20].

Similarly, a study from Ethiopia in Wukro and Butajira districts came up with similar findings indicating that about 75% of mothers had non-institutional childbirths [21]. These significant differences in our study results from the past studies can be explained by the fact that the previous studies were mainly conducted with an urban population only, unlike our study, which gives information on rural and urban people across all the Regional State districts. The higher rates of institutional childbirths amongst the inner-city (urban) women have been due to their better access to health education and information (health promotion programs work to the urban residents’ advantage). Therefore, a preference for institutional childbirths could be seen. Moreover, less travel time and better transportation facilities in the urban areas might have played an essential role for mothers residing near healthcare institutions, enabling them to better access maternal healthcare services compared to those living in rural and remote settings. Distance from healthcare facilities has been a critical factor that significantly influenced mothers not to choose ICBs. Our study’s findings closely related to some of the descriptive studies conducted in Ethiopia’s rural districts, which stated that large numbers of mothers opted for NICBs with the assistance of traditional birth attendants, more than 80% of mothers had non-institutional childbirths in the rural areas, e.g., in the northwest, southeast, and middle of the country) [20,22].

A large proportion of the mothers who have had childbirths non-institutionally, once they develop obstetric complications after the failure of traditional interventions, tend to seek help from skilled healthcare personnel, demonstrating the lack of understanding regarding the seriousness of health problems [20,22]. Maternal age influences health-seeking behavior, as it has been found to be associated with maternal healthcare utilization (e.g., the rate of childbirth in healthcare institutions) due to the difference in the younger and older mothers’ experience level. In general, older mothers usually prefer their childbirths outside of institutions. At the same time, younger women are more likely to accept modern health care since they have comparatively better education and more exposure to information. Women with better education are more careful about their pregnancies. Therefore, they tend towards skilled professionals, while the mothers with no or minimal formal education and elderly rely mostly on their experience (accumulated cultural knowledge from previous pregnancies and births) and believe that modern healthcare is barely essential [22,23].

Our study highlighted a few more factors involved in the low institutional childbirth service utilization, such as the distance from the healthcare facility, hesitation towards a male midwife, prolonged transportation time for mothers to the healthcare facilities, fear of low skilled health worker, non-involvement of mothers in key decision making, high cost for referral via ambulance and absence of childbirth services in many of the healthcare facilities. The absences of services in the childbirth institutions also reflected the behavior of health workers who were not committed to their work and broke mothers’ trust in these facilities. Similar reasons counted in a few other studies done in Zimbabwe and Namibia, indicating the primary reasons women do not use institutional childbirth services, like cultural practices, individual perceptions, and male health workers’ presence during childbirth being culturally offensive in most parts of Africa [24,25]. A study on factors contributing to the low uptake of specialized care during birth in Malindi, Kenya, identified distance from the healthcare facility, lack of transportation, and repulsion from a male midwife as significant barriers to pursuing skilled health care for their childbirths [26]. During childbirth, men’s presence is considered invasive, but, in some ethnic groups, male family members are not supposed to occupy the same physical space as the mother for many days, which may add tension and lead to family and social disorder [26]. The mothers’ marital status and occupation were statistically significantly associated with childbirth services utilization in our study. More married and working mothers preferred institutional childbirths than mothers who were not married and non-employed. This result has shared similarity to some of the previous studies in Ethiopia [22,27,28,29]. Ethnicity was another variable that showed a statistically significant association with non-institutional childbirths. Mothers from the Nuer ethnic group and Majang ethnic group were two times more likely to have non-institutional childbirths than mothers from other ethnic groups. This discovery is consistent with the previous studies conducted on institutional childbirth utilization in Tanzania [30]. Selection bias was minimized since it was a community-based study with a probability sampling technique. Data collectors were non-healthcare workers, and they were unaware of the desired answers. We included all districts, rural and urban areas with functional healthcare facilities, and adequate sample size. All participants included in the study responded to the questionnaire.

There were some weaknesses as well, which need to be mentioned. This study did not include the population where there was no functional healthcare facility that might have affected its findings’ generalization to some extent. The nature of the cross-sectional survey does not allow for causality for the effect of the relationship. Qualitative methods, focus group discussions on exploring more mothers’ opinions, and in-depth reasons for not utilizing institutional birth services were not employed. As the study collected retrospective information from the mothers on having childbirth in the last two years, the possibility of recall bias and misreporting of events cannot be excluded. Sampling bias may be possible due to the convenient sampling technique used to select five villages in each district by considering the estimated number of births in the village populations. Finally, Our study did not examine all pregnancy outcomes; we only interviewed women with a child below two years because miscarriage and stillbirth were challenging to find or interview the mothers with such pregnancy results. Thus, the perinatal outcome analyses, including gravidity and parity, were unable to be conducted.

## 5. Conclusions

This study confirmed that most of the births were non-institutional without skilled attendants two years before the survey. This study provided novel information on childbirth practices in the state, which can add some value for the public health managers and implementers to improve maternal and child health in the country and region.

The most significant socio-demographic factors associated with childbirth service utilization were; occupation, marital status, and ethnicity. Future studies need to explore the risks associated with non-institutional childbirths, particularly for ethnic groups who were less likely to undergo institutional childbirths. Public health professionals, clinicians, governmental officials, and community leaders should increase the institutional birth rates to reduce maternal mortality and morbidity and control risk factors linked with non-institutional childbirths. Raise community awareness on the risk of non-institutional delivery. Improving education among females beyond primary school needs to be strongly encouraged to improve mothers’ education Increase the mother’s autonomy within the family to enhance their decision-making on their maternal health. Raising awareness of pregnancy-related complications and the utilization of healthcare facilities in community awareness programs must focus on the danger signs surrounding pregnancy and childbirth. Mothers should be educated towards the modern culture of institutional birth and more open to accepting maternal healthcare services. Increase the training of female midwife professionals to avoid male midwives playing a role in handling childbirth, thus the mothers should get assistance with the same gender. The health administration teams should ensure support to healthcare providers through constant healthcare equipment supplies, supervision, monitoring, and evaluation to influence their attitude to improve service quality and increased health for institutional childbirth.

## Figures and Tables

**Figure 1 ijerph-18-02859-f001:**
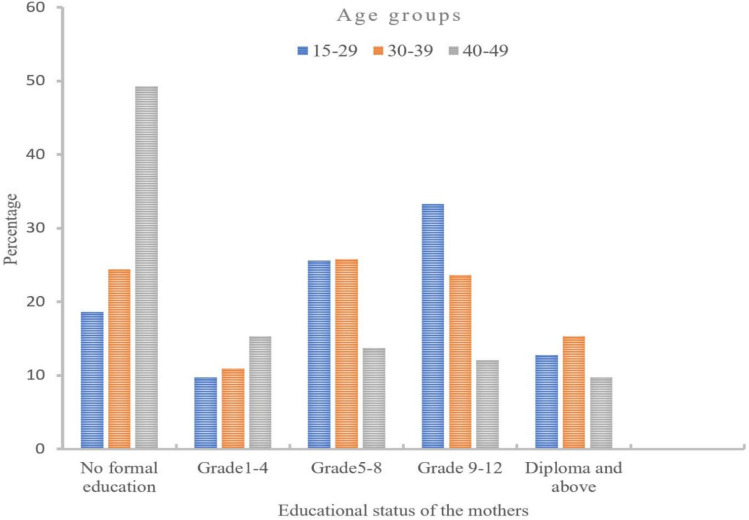
Respondents educational status by age groups.

**Figure 2 ijerph-18-02859-f002:**
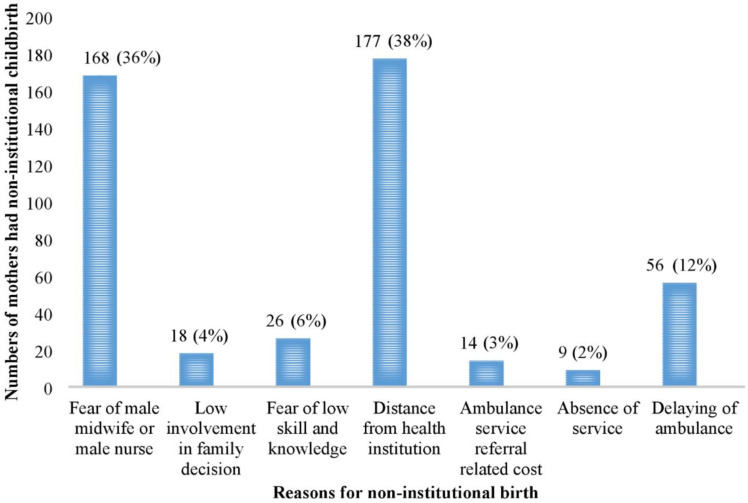
Respondents’ reasons for their non-institutional childbirths.

**Table 1 ijerph-18-02859-t001:** Women’s characteristics and association with institutional and non-institutional childbirths.

Variable	All Respondents		Non-Healthcare Institutional Birth		Healthcare Institutional Birth		*χ* ^2^	*p*-Value
	Number	Percent	Number	Percent	Number	Percent		
Age groups								
15–29	258	39.0	178	69.0	80	31.0	1.241	0.538
30–40	275	41.9	198	72.0	77	28.0		
40–49	124	18.9	92	74.2	32	25.8		
Marital Status								
Married	531	80.8	365	68.7	166	31.3	8.409	0.015
Single	60	9.0	49	81.7	11	18.3		
Others *	66	10.0	54	81.8	12	18.2		
Occupation								
Housewife	387	58.9	287	74.2	100	25.8	13.373	0.004
Employee	93	14.2	52	55.9	41	44.1		
Merchant	42	6.4	33	78.6	9	21.4		
Others **	135	20.5	96	71.1	39	28.9		
Educational status								
No formal education	176	26.8	139	79	37	21.0	14.306	0.006
Grade 1–4	74	11.3	56	75.7	18	24.3		
Grade 5–8	154	23.4	108	70.1	46	29.9		
Grade 9–12	166	25.3	115	69.3	51	30.7		
Diploma and above	87	57.5	50	57.5	37	42.5		
Religion								
Christian	621	94.5	444	71.5	177	28.5	0.453	0.797
Muslim	13	2.0	9	69.2	4	30.8		
Others ***	23	3.5	15	65.2	8	34.8		
Monthly income								
Below 1000 ETB	426	64.8	314	73.7	112	26.3	5.348	0.148
1001–3000 ETB	155	23.6	105	67.7	50	32.3		
3001–6000 ETB	64	9.7	43	67.2	21	32.8		
6001 ETB and above	12	1.8	6	50.0	6	50.0		
Ethnicity								
Anywaa	235	35.8	156	66.4	79	33.6	16.795	0.005
Nuer	226	34.4	178	78.8	48	21.2		
Majang	87	13.2	67	77.0	20	23.0		
Komo	8	1.2	6	75.0	2	25.0		
Opuo	10	1.5	5	50.0	5	50.0		
Others ****	91	13.9	56	61.5	35	38.5		

Abbreviation: ETB, Ethiopian Birr (currency). * Others marital status: Divorced, widowed, never married, illegal partner. ** Others Occupation: Famer, Daily laborer, upon. *** Others Religion: Those who have no religion to believed and traditional believers. **** Others Ethnicity: All ethnic groups from another part of the country.

**Table 2 ijerph-18-02859-t002:** Plan for the place of birth service utilization.

Variable	Place of Birth				*χ* ^2^	*p*-Value
	Institutional Birth		Non-institutional Birth			
	Number	Percent	Number	Percent		
Plan for the place of delivery						
Yes	160	41	230	59	70.375	0.006
No	29	10.9	238	89.1		
For yes where to delivery						
Non-institutional birth	1	1.7	57	98.3	43.496	<0.001
Institutional birth	159	47.9	173	52.1		

**Table 3 ijerph-18-02859-t003:** Socio-demographic factors associated with institutional and non-institutional childbirths.

Variable	Place of Birth		OR (95 % CI)	*p*-Value	AOR (95 % CI)	*p*-Value
	Non-Institutional Birth *n* (%)	Institutional Birth *n* (%)				
Age groups						
15–29	178 (69.0%)	80 (31.0%)	0.77 (0.47–1.25)	0.296		
30–40	198 (72.0%)	77 (28.0%)	0.89 (0.55–1.44)	0.649		
40–49	92 (74.2%)	32 (25.8%)	1			
Marital Status						
Married	365 (68.7%)	166 (31.3%)	0.49 (0.25–0.93)	0.031	0.36 (0.17–0.74)	0.006
Single	49 (81.7%)	11 (18.3%)	0.99 (0.40–2.44)	0.982	1.36 (0.53–3.47)	0.524
Others *	54 (81.8%)	12(18.2%)	1		1	
Occupation						
Housewife	287 (74.2%)	100 (25.8%)	0.47 (0.69–0.91)	0.039	1.87 (1.14–3.05)	0.012
Employee	52 (55.9%)	41 (44.1%)	0.52 (0.29–0.89)	0.019	0.57 (0.31–0.82)	0.029
Merchant	33 (78.6%)	9 (21.4%)	1.49 (0.65–3.40)	0.344	1.44 (0.61–3.37)	0.397
Others **	96 (71.1%)	39 (28.9%)	1		1	
Educational status						
No formal education	139 (79.0%)	37 (21.0%)	2.78 (1.59–4.86)	<0.01	2.49 (1.40–4.41)	0.002
Grade 1–4	56 (75.7%)	18 (24.3%)	2.30 (1.16–4.54)	0.016	2.15 (1.08–4.27)	0.029
Grade 5–8	108 (70.1%)	46 (29.9%)	1.74 (1.00–3.00)	0.048	1.77 (1.01–3.08)	0.045
Grade 9–12	115 (69.3%)	51 (30.7%)	1.67 (0.97–2.85)	0.062	1.74 (1.00–3.01)	0.049
Diploma and above	50 (57.5%)	37 (42.5%)	1		1	
Religion						
Christian	444 (71.5%)	177 (28.5%)	1.34 (0.55–3.21)	0.515		
Muslim	9 (69.2%)	4 (30.8%)	1.20 (0.27–5.15)	0.806		
Others ***	15 (65.2%)	8 (34.8%)	1			
Monthly income						
Below 1000 ETB	314 (73.7%)	112 (26.3%)	2.80 (0.88–8.87)	0.079		
1001–3000 ETB	105 (67.7%)	50 (32.3%)	2.10 (0.64–6.83)	0.218		
3001–6000 ETB	43 (67.2%)	21 (32.8%)	2.05 (0.58–7.11)	0.26		
6001 ETB and above	6 (50.0%)	6 (50.0%)	1			
Ethnicity						
Anywaa	156 (66.4%)	79 (33.6%)	1.23 (0.74–2.03)	0.411	1.24 (0.74–2.07)	0.408
Nuer	178 (78.8%)	48 (21.2%)	2.32 (1.36–3.93)	0.002	2.12 (1.23–3.63)	0.006
Majang	67 (77.0%)	20 (23.0%)	2.09 (1.08–4.02)	0.027	1.98 (1.02–3.83)	0.043
Komo	6 (75.0%)	2 (25.0%)	1.88 (0.35–9.81)	0.457	1.49 (0.28–7.95)	0.636
Opuo	5 (50.0%)	5 (50.0%)	0.63 (0.16–2.31)	0.482	0.57 (0.15–2.14)	0.404
Others ****	56 (61.5%)	35 (38.5%)	1		1	

Abbreviation: ETB, Ethiopian Birr. ** Others marital status: Divorced, widowed, never married, illegal partner. ** Others Occupation: Famer, daily laborer, upon. *** Others Religion: Those who have no religion to believed and traditional believers. **** Others Ethnicity: All ethnic groups from another part of the country.

## Data Availability

All data produced or analyzed throughout this study included in this article is attached as additional files.

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
