# Peer review of "Non-Institutional Childbirths and the Associated Socio-Demographic Factors in Gambella Regional State, Ethiopia"

_ijerph, 2021, doi:10.3390/ijerph18062859_

Round 1
Reviewer 1 Report
Please check the data accuracy in Table3.
- In the column of OR (95%CI), the data of housewife presented here is 1.17 (0.75-0.180), please check this data. This should be incorrect.
- In the column of OR, the data of no formal education, the p-value showed 0. Please consider to revise it to p<0.01.
- In the column of AOR (95%CI), the aOR data of employee showed 0.57 (0.31-0.82). In my opinion, the p-value should be less than 0.05. Please clarify or revise this data.
- Please check the data accuracy in your manuscript again.
Author Response
Please check the data accuracy in Table3.
Reply: - All the data in the table 3 have been revised
- In the column of OR (95%CI), the data of housewife presented here is 1.17 (0.75-0.180), please check this data. This should be incorrect.
Reply: - This data was checked, revised, and corrected to 0.47 (0.69-0.091).
- In the column of OR, the data of no formal education, the p-value showed 0. Please consider revising it to p<0.01.
Reply: - It was revised to P <0.001.
- In the column of AOR (95% CI), the AOR data of employee showed 0.57 (0.31-0.82). In my opinion, the p-value should be less than 0.05. Please clarify or revise this data.
Reply: - It was revised P-value 0.029.
- Please check the data accuracy in your manuscript again.
Reply: - The data accuracy in the manuscript have been checked and the errors have been revised.

Reviewer 2 Report
It is a pleasure to read this revised paper. This is an improved manuscript in many respects. However, the English language (style and grammar) should be revised. Additionally, justification for biases should be part of the discussion section.
Author Response
Comments and Suggestions for Authors
It is a pleasure to read this revised paper. This is an improved manuscript in many respects. However, the English language (style and grammar) should be revised. Additionally, justification for biases should be part of the discussion section.
Reply: - The English language (style and grammar) of the entire manuscript has been revised and justification for biases has been included into the discussion part.

This manuscript is a resubmission of an earlier submission. The following is a list of the peer review reports and author responses from that submission.
Round 1
Reviewer 1 Report
To the authors,
The study results illustrated the current childbirth culture and practice in Ethiopia and also highlighted the low institutional childbirth rate in the Gambella regional state, which may be related to the higher maternal mortality and morbidity rates in this developing country. The authors provided a well-structured study design to investigate the reasons not to give birth in an institution and the results could help health professionals to improve child and maternal health in the future.
I have some comments and suggestions for the authors and hope that my comments are constructive to them.
Abstract
- Line 29. “Socio-demographic 29 factors associated with NICB were: employee (Adjusted Odds Ratio (AOR) = Nuer (AOR = 2.12, 95 30 % CI: 1.23 - 3.63)..”This sentence is unclear. Please revised it.
Introduction
- Line 45. “the world’s the developed countries and ..” This sentence is unclear. Please revised it.
- the introduction part is well-structured and the readers could understand the importance of this topic and also increase the readability of this manuscript.
Methods
- For the perinatal outcome analysis, the number of previous delivery babies (gravidity and parity) is an important factor. Although in your questionnaire, you didn’t collect the data of gravidity and parity. It is still recommended that the authors include the gravidity and parity for discussion. How about the correlation between gravidity and maternal age or education?
Results
- Table 2 and Table 4. it is recommended to show the percentage of each category in the column itself.
- Figure 2. The “Total” bar is useless, please remove it.
Discussion
- The data of “gravidity and parity” is important for perinatal outcome analysis, please add this to the limitation part and add some discussion about it.
- The study results showed that the most significant factors were occupation, marital status, and ethnicity. However, in table 3, the educational status also seems to be significantly associated with the non-institutional childbirth. How to interpret this educational status factor? Could it be highly associated with occupational factors? How to well differentiate them?
- Since that the authors reported these socio-demographic factors were so important for non-institutional childbirth, please add some suggestions to improve the willingness to give birth in the hospital or maternal clinics for pregnant women in Ethiopia. For example, the mobile medical care in the rural area or public health education, or any policies for this topic now conducting in Ethiopia?
Author Response
We are happy to receive the comments from reviewers. Meanwhile, this is an encouragement for us to do such studies further. We carefully revised the manuscript according to the comments. We made a response to all comments point-by-point and pointed to the revised place. The response named "Reply to Reviewer Comments "as a separate file.
We confirm the authors' list as follows (not changed the list):
Gnkambo Agwa Gora, Muhammad Farooq Umer, Peter Obang Ojulu, Sintayehu Tsegaye Betaw, Akwoma Okugn Cham, Ojulu Agwa Gora, Xin Qi*
Moreover, the corresponding authors as follows:
Corresponding author: Xin Qi (xin.qi@mail.xjtu.edu.cn), Department of Epidemiology and Biostatistics, School of Public Health, Xi'an Jiaotong University, Xi'an, China
Sincerely,
Gnkambo Agwa Gora (goragnkambo@gmail.com,/goragnkambo@outlook.com),
1, Gambela People National Regional State Bureau of Health
Gambela People National Regional State. P.O.Box 142, Gambella 5440, Ethiopia
2, School of Public Health,
Xi’an Jiaotong University,
Xi'an, Shaanxi 710061, China.

Reviewer 2 Report
The authors aimed to assess the prevalence of non-institutional childbirths in Gambella Regional State, Ethiopia, and to identify socio-demographic factors responsible for non-utilisation of available birth services by the reproductive-aged mothers.
In general, this is a potentially stimulating article. I have some suggestions for your manuscript.
The English language (style and grammar) should be revised, many typos, syntax and grammatical errors through the paper are found.
Keywords: Could add to the title and abstract. Institutional birth is misspelled.
- Background: The rationale for the study was explained. However, your introduction could better inform the reader why the paper is important in terms of the relation between birth quality of care, gender inequalities, and women’s empowerment, and what is original about the present research. The authors should demonstrate what new knowledge is brought by their study (and originality comparing to the existing studies concerning the utilization of institutional delivery and associated factors in Ethiopia) and its level of interest for an international readership. What is known about utilization of institutional delivery and persistent intersectional inequities in low-income countries? Why did the authors focus their analyses only on the socio-demographic characteristics of mothers and not on the decision-making process regarding the delivery and obstetric characteristics of mothers?
The two last sentences of the introduction should be moved to the discussion section.
- Methods: Methods need to be described with sufficient detail.
2.1. Study Design: Collected variables need to be described and more detail concerning the data collection process is needed to clarify eligibility criteria (inclusion and exclusion of participants) and assessment, and how were eligible participants invited to participate. Please specify random sampling technique used to select the participants.
2.2. Study Area: The description of the study area would benefit from inclusion of variables of interest, such as sociodemographic variables.
2.3. Data Analysis: Analyses would benefit from being specified and presented linked to the aims.
- Results: Table 1 could be merged to table 2. Important data concerning previous births, delivery plans, and antenatal care is missing. Also, information concerning the characterization of the non-institutional and institutional deliveries would be very important given that giving birth in a health facility does not necessarily equate with high-quality care or fewer maternal deaths. Figure 1 does not add crucial information. Concerning figure 2, how was the question constructed, was it the main reason and was it applied just to the participants that with non-institutional childbirth? Those variables were crucial to be more deepen evaluated. It seems that some of the reasons indicate a willingness to an institutional birth. Also, additional analyses concerning childbirth plans is missing. Please indicate which was the reference category for place of birth (dependent variable).
- Discussion: Please comment on the very unusual 100% participation rate. If the main reason for different findings form previous research is the inclusion of people form rural areas, the authors should present information considering the place of residence of the participants and identify its correlation to place of birth. Also, the authors discuss findings related to non-collected/presented results. Discussion needs to focus on delivery intersectional inequities.
- Strengths and Limitations of the Study: Strengths of the study should be revised, namely justification for bias.
- Conclusion: Recommendations for safe delivery care policy should be included.
References: Need to be completed and updated. Consider including, for example:
Anshebo D, Geda B, Mecha A, Liru A, Ahmed R (2020) Utilization of institutional delivery and associated factors among mothers in Hosanna Town, Hadiya Zone, Southern Ethiopia: A community-based cross-sectional study. PLoS ONE 15(12): e0243350. https://doi.org/10.1371/journal.pone.0243350
Supon Limwattananon, Viroj Tangcharoensathien & Supakit Sirilak (2011)Trends and inequities in where women delivered their babies in 25 low-income countries: evidence from Demographic and Health Surveys, Reproductive Health Matters, 19:37, 75-85, DOI: 10.1016/S0968-8080(11)37564-7
Author Response

(The authors gave the same response as above.)
